# Molecular Characterization of the First African Swine Fever Virus Genotype II Strains Identified from Mainland Italy, 2022

**DOI:** 10.3390/pathogens12030372

**Published:** 2023-02-24

**Authors:** Monica Giammarioli, Dondo Alessandro, Cesare Cammà, Loretta Masoero, Claudia Torresi, Maurilia Marcacci, Simona Zoppi, Valentina Curini, Antonio Rinaldi, Elisabetta Rossi, Cristina Casciari, Michela Pela, Claudia Pellegrini, Carmen Iscaro, Francesco Feliziani

**Affiliations:** 1Istituto Zooprofilattico Sperimentale Umbria e Marche “Togo Rosati”, 06126 Perugia, Italy; 2Istituto Zooprofilattico Sperimentale del Piemonte, Liguria e Valle d’Aosta, 10154 Torino, Italy; 3Istituto Zooprofilattico Sperimentale dell’Abruzzo e del Molise “G. Caporale”, Campo Boario, 64100 Teramo, Italy

**Keywords:** African swine fever virus (ASFV), genotype II, ASF outbreak, mainland Italy, wild boar population, next-generation sequencing (NGS), phylogenetic analysis

## Abstract

African swine fever (ASF) is responsible for important socio-economic effects in the global pig industry, especially for countries with large-scale piggery sectors. In January 2022, the African swine fever virus (ASFV) genotype II was identified in a wild boar population in mainland Italy (Piedmont region). This study describes the molecular characterization, by Sanger and next-generation sequencing (NGS), of the first index case 632/AL/2022 and of another isolate (2802/AL/2022) reported in the same month, in close proximity to the first, following multiple ASF outbreaks. Phylogenetic analysis based on the B646L gene and NGS clustered the isolates 632/AL/2022 and 2802/AL/2022 within the wide and most homogeneous p72 genotype II that includes viruses from European and Asian countries. The consensus sequence obtained from the ASFV 2802/AL/2022 isolate was 190,598 nucleotides in length and had a mean GC content of 38.38%. At the whole-genome level, ASF isolate 2802/AL/2022 showed a close genetic correlation with the other representative ASFV genotype II strains isolated between April 2007 and January 2022 from wild and domestic pigs in Eastern/Central European (EU) and Asian countries. CVR subtyping clustered the two Italian ASFV strains within the major CVR variant circulating since the first virus introduction in Georgia in 2007. Intergenic region I73R-I329L subtyping placed the Italian ASFV isolates within the variant identical to the strains frequently identified among wild boars and domestic pigs. Presently, given the high sequence similarity, it is impossible to trace the precise geographic origin of the virus at a country level. Moreover, the full-length sequences available in the NCBI are not completely representative of all affected territories.

## 1. Introduction

African swine fever (ASF), which is caused by a large enveloped linear double-stranded DNA virus [1], is responsible for important socio-economic consequences in the global swine industry, especially for countries with wide-scale pig production.

It was first identified in Kenya in 1921 [2], and prior to 2007 was mainly endemic in Africa. After a genotype I pandemic wave started in the Iberian Peninsula in 1960, the virus was eradicated worldwide, although it remained endemic in Sardinia Island [3]. In 2007, ASF was introduced to the Republic of Georgia through the port of Poti, likely via improperly disposed waste from international ships carrying contaminated pork or pork products, which were then used as swine feed [4]. Subsequently, the genotype II virus spread to Armenia and the Russian Federation. Afterward, the virus invaded the European Union in 2014 [5]. Later, in 2018, it was first confirmed in China [6] following several outbreaks in Siberia in 2017 [7]. Since then, the disease has rapidly spread to 15 other Asian and Pacific countries [8]. In early 2020, it was also reported in two North-Eastern states of India [9]. The countries most recently affected include Papua New Guinea, Germany [10], the Dominican Republic [11] and Haiti [12].

ASF genotype I is still considered an endemic in Sardinia, although only sporadic indirect traces of the virus are currently detectable. In Italy, two ASF virus (ASFV) introductions in pigs have been reported in the past. The first was reported in 1967 in 28 Italian provinces. The second incursion was reported in Piedmont in March 1983 and represented the unique escape of ASFV from Sardinia; wild boar (*Sus scrofa*) meat imprudently imported from Sardinia has been incriminated as the cause of this outbreak. Strict quarantine and slaughter measures confined the spread of the disease, and the outbreaks were successfully eradicated. Recently, in 2022, ASFV was detected in a wild boar in mainland Italy (Piedmont region) [13].

ASFV is the only member of the family *Asfarviridae*, genus *Asfivirus*. The genome is approximately 170–190 kb in size and is divided into the left variable region (LVR, 38–48 kb), the central conserved region (CCR, approximately 125 kb) and the right variable region (RVR, 13–22 kb) [14,15]. There is a central variable region (CVR) of approximately 400 bp [16] within the CCR. The nature of CVR variation and the genetic mechanisms involved are unknown. The molecular basis of this variation includes alterations in the number and type of tandem repeated amino acid tetramers within a late viral gene, 9-RL. Analysis of the B602L amino acid sequences reveals a high degree of variability among different ASFV isolates [17,18,19,20,21,22].

Differences in the genome length are largely caused by the gain or loss of members of the five multigene families (MGF 100, 110, 300, 360 and 505/530) flanking the variable regions [23]. LVR and RVR can have a variable number of MGF genes. Variations within these regions are observed during the viral adaptation to monkey cell lines [24] and appear to be associated with reduced virulence [25]. Furthermore, minor length variations are associated with the number of tandem repeats located at loci within the coding regions and in intergenic regions (IGR) [26]. These variable regions are important for the evolutionary analysis of ASFV.

Molecular characterization of ASFV during outbreaks is significant for investigating the virus origin, quickly differentiating between closely related strains [27] and extening our knowledge of the molecular evolution of the virus and its epidemiology. The most common approach for the genotyping of ASFV during outbreaks is based on analyzing the C-terminal end of the B646L gene, which encodes the p72 capsid protein [28]. This approach has allowed for the identification of 22 distinct p72 genotypes (I-XXII) among virus strains from Eastern and Southern African countries [29]. ASFV genotype XXIII was identified in Ethiopia and was found to be derived from the same evolutionary branch as the IX and X genotypes prevalent in East African countries and the Democratic Republic of Congo [30]. The genotype XXIV has been identified in soft tick samples collected from Gorongosa National Forest Park, Mozambique [30]. The E183L gene, encoding the p54 protein and the CVR within the B602L gene, was also sequenced to distinguish between geographically and temporally constrained p72 genotypes [17,18,19,20,21,22].

Furthermore, to evaluate molecular differences between the ASFV strains, especially the evolutionary trend of strains from the same region, other approaches include the assessment of the: p30-encoding gene (CP204L) [31,32,33]; TRS within the IGR between I73R and I329L [32,33]; CD2v-encoding gene (EP402R) [33]; thymidine kinase (TK) gene [34]; J268L, Bt/Sj, KP86R [22] and O174L genes [35], and the C315R/C147L region [36]. Sequence analysis showed that the different isolates have partial differences in the length and sequences of the J268L, Bt/Sj and KP86R genes, which may be used to distinguish between evolutionarily similar isolates [22].

The recent innovations in whole-genome sequencing can facilitate comprehensive genotyping and provide data that are essential for elucidating the biology and genetic characteristics of ASFV. The use of next-generation sequencing (NGS) and bioinformatics analysis for detecting and identifying ASFV from clinical samples isolated from outbreaks have been previously described [37,38,39,40,41].

In this study, we used Sanger sequencing and NGS to investigate the epidemiological links between the ASFV strains causing outbreaks amongst wild boars in mainland Italy in January 2022.

## 2. Materials and Methods

We analyzed bone marrow and spleen samples from the first index case (632/AL/2022) and one other virus (2802/AL/2022) collected from wild boars in the affected area during the outbreaks which occurred in Italy (Piedmont region) in the first two weeks of January 2022 [13]. The samples were tested at the National Reference Laboratory (CEREP) in a Biosafety Level 3 (BSL3) facility for diagnostic confirmation.

### 2.1. Samples Preparation

The ASFV was isolated from the positive samples as described in the *OIE Manual of Diagnostic Tests and Vaccines for Terrestrial Animals*, Chapter 3.9.1, 2021. The strains were haemadsorbed in pig macrophage cultures. The ASFV purification was performed as previously described by Enjuanes et al. [42], with some modifications. Briefly, the virus suspension was treated at 37 °C for 90 min with an RNase Cocktail (5 U and 200 U of RNase A and RNase T1, respectively, Thermo Fisher Scientific: Baltic UAB, Vilnius, Lithuania) and 150 U Turbo DNase (Thermo Fisher Scientific: Baltic UAB, Vilnius, Lithuania). This was followed by treatment with trypsin (6 mg/mL) for 2.5 h at 37 °C and subsequent ultracentrifugation in a 5–20% sucrose gradient in phosphate buffer saline (PBS) at 4 °C for 90 min in a swing rotor (Sw 41) at 68,600× *g* (Beckman Coulter Inc., Brea, CA, USA). The original clinical materials (spleen, bone marrow, etc.) were subjected to three/four freeze–thaw cycles to homogenize the samples. After centrifugation at 6000× *g* for 10 min at 4 °C, the supernatants were filtered through 0.45 μm syringe filters. The filtered homogenates were treated at 37 °C for 90 min with the RNase Cocktail and Turbo DNase as previously described [41].

### 2.2. Partial Sequencing (Sanger)

The samples were genotyped by partial sequencing (Sanger) of a fragment of the B646L gene (p72), a fragment of the B602L gene (CVR) and tandem repeat sequences located between the I73R and I329L genes (ECORI), using previously described primers and thermal cycling conditions (refs) shown [33]. Viral DNA was extracted from the lysed samples and from the ASF isolates using the High Pure DNA nucleic acid kit (Roche Diagnostics GmBH: Mannheim, Germany) following the manufacturer’s instructions. The selected positive samples were characterized by partial sequencing (Sanger) on an ABI PRISM 3130 Genetic Analyser (Applied Biosystems: Waltham, MA, USA). Bidirectional sequencing was performed for each of the gene targets, and three independent reactions were performed for each sample. The sequences were then analyzed using the DNAStar package v.15 [43]. The nucleotide sequences were aligned using Clustal X with the ASF reference strains retrieved from PubMed at the National Center for Biotechnology Information (NCBI) (http://www.ncbi.nlm.nih.gov/ accessed on 12 December 2022). The manual sequence editing was performed using BioEdit software (version 7.0.) [44], and phylogenetic analysis was performed using MEGA v. 7 [45] with the GTR model of sequence evolution and gamma distribution (GTR+G+I). The robustness of the clusters was tested by performing 10,000 bootstrap replicates: branches with bootstrap values below 70% are not shown [45].

### 2.3. Shotgun Metagenomic Analysis

The spleen sample of 2802/AL/2022 was selected for shotgun metagenomic analysis. The DNA samples were quantified using the Qubit^®^ DNA HS Assay Kit (Thermo Fisher Scientific: Waltham, MA, USA) and then used for library preparation with Illumina^®^ DNA Prep, (M) Tagmentation (96 samples) (Illumina Inc.: San Diego, CA, USA) according to the manufacturer’s protocol. Deep sequencing was performed on the NextSeq500 (Illumina Inc.: San Diego, CA, USA) using the NextSeq 500/550 Mid Output Reagent Cartridge v2 (300 cycles) (Illumina Inc.: San Diego, CA, USA) and standard 150 bp paired-end reads. After quality checking and trimming of the raw reads data using FastQC v0.11.5 and Trimmomatic v0.36, respectively, host depletion was performed using Bowtie2 [46]. The reads were mapped using the BWA software package v.0.7.17 [47], with the ASFV Georgia 2007/1 sequence (accession number FR682468.2) as a reference. The iVar v1.3.1 tool was used to define a consensus sequence based on the mapping results [48].

Genome annotation was performed as previously described [37], using the GATU software [49] with the ASFV Georgia 2007/1 sequence (accession number FR682468.2) as a reference. The annotations were manually verified and curated using the Ugene software package [50].

A maximum likelihood (ML) phylogenetic tree was constructed using the Tamura–Nei parameter model in MEGA v.7 with 10,000 bootstrap replicates [45].

## 3. Results

Two representative isolates (632/AL/2022 and 2802/AL/2022) from the outbreak area in Northwest Italy were selected for molecular characterization.

The sample 632/AL/2022 was genotyped using partial sequencing (Sanger) of the B646L gene (p72), the B602L gene (CVR) and a TRS located between the I73R and I329L genes. The nucleotide sequences from the 478 bp region of the p72 genes were analyzed and compared with the previously defined reference sequences retrieved from GenBank. The phylogenetic analysis established that sample 632/AL/2022 was part of the largest and most homogeneous p72 genotype II cluster, which also includes strains from European and Asian countries (Figure 1). The sequence similarity ranged from 92.40 to 99.60% between the various subtypes. The phylogenetic tree is shown in Figure 1. It was not possible to carry out a complete genome sequence of the first index case (632/AL/2022) by NGS, as the starting material, represented by the bone marrow, was very poor.

The sample 2802/AL/2022 was selected for full–genome characterization by NGS. The sequence obtained showed a horizontal coverage of 100.00%, compared with that of the reference genome ASFV Georgia 2007 sequence (FR682468.2) and a mean vertical coverage of 885,*45x*. The consensus sequence of ASFV 2802/AL/2022 was 190,598 nucleotides in length and had a mean GC content of 38.38%. It contained 192 ORFs, encoding for structural–functional and uncharacterized proteins. The complete sequence was analyzed using BLASTn to identify the closest sequence match in the GenBank database. The whole-genome sequence of the 2802/AL/2022 ASFV positive sample showed a close genetic correlation with the other representative ASFV genotype II strains isolated between April 2007 and January 2022 from wild boars and domestic pigs in Eastern/Central European (EU) and Asian countries (Figure 2).

Additionally, in order to trace the outbreaks/cases, the genetic characterization of the two isolates was conducted using the EURL-genotyping standardized procedures.

A polymerase chain reaction (PCR) targeting the B602L variable region yielded a ~180 bp fragment from the two cases analyzed. Analysis of the amino acid tetramer repeat sequences within the CVR of the two ASFV isolates in this study revealed the presence of nine repeats identical to those identified among genotype II isolates from European and Asiatic countries (Figure 3). Comparison of the deduced amino acid sequences derived from the variable region of the B602L with 51 genotype II reference strains from European and Asian countries isolated between 2007 and 2022 identified 10 unique sequences of amino acid tetramers (BNDBNDBNAL) (Figure 3). Therefore, CVR subtyping clustered these ASFV isolates detected in Italian wild boar within the CVR variant majority circulating in the EU and Asian countries since their first introduction in Georgia in 2007.

To further define the most likely origin of the introduction of the ASFV genotype II in Italy, we compared TRS in the IGR between I73R and I329L with the representative ASFV genotype II strains. Intergenic region I73R-I329L subtyping clustered the Italian ASFV isolates with variants that have been frequently identified among wild boar and domestic pigs in EU countries since ASFV’s introduction in 2014 and after 2018 in Asian countries (Figure 4).

The partial sequences and the complete genome sequence have been deposited in the NCBI (ON108572, ON108573, ON108574 and ON108571).

## 4. Discussion

To the best of our knowledge, Italy is the only country in Europe where the ASFV genotypes I and II coexist. In Sardinia, the ASFV genotype I was first identified in 1978 in the Cagliari province, presumably introduced from the Iberian Peninsula via food waste, which was subsequently fed to pigs [51]. The geographical location of the infection on an island has helped to limit the disease spread in the peninsula. In fact, in 40 years, only two introductions of the virus in mainland Italy have been reported. The first ASFV genotype II introduction in the mainland regions occurred in a wild boar, detected in Northwest Italy in January 2022 [13]. This study represents a comprehensive attempt to resolve the intragenotypic relationships of the two genotype II isolates that are geographically and temporally linked to causing outbreaks in mainland Italy in January 2022 and to detect epidemiological links that may exist between outbreaks in previously affected countries. The molecular characterization of the different ASFV outbreaks is crucial for investigating the introduction of the virus and for quickly differentiating between closely related strains [27] to extend our knowledge of ASFV’s viral evolution and epidemiology. Understanding the molecular evolution of ASFV isolates is necessary for applying effective prevention and control strategies.

The combined p72, ECORI and CVR approach and full–genome strategy have been used to achieve high levels of discrimination among closely related virus isolates. The two Italian strains, 632/AL/2022 and 2802/AL/2022, were defined as genotype II based on partial p72 gene and full–genome sequencing. The nucleotide sequences obtained from the two isolates revealed 100% similarity with the earlier characterized genotype II reference strains available in GenBank, and we have not identified any subgroups within genotype II (Figure 1 and Figure 2). The B602L gene of ASFV is a hypervariable genetic marker that has shown useful for high-resolution discrimination of viruses that are identical according to their p72 and p54 genotypes. This region contains 12 base-pair repeats encoding four amino acids that vary in number and sequence when the genomes of different isolates are compared [26]. The CVR subtyping clustered the Italian wild boar ASFVs recently detected in Italy within the CVR variant I, with the majority circulating in Eastern European countries and Asian countries since the first introduction of the virus to Georgia in 2007. In addition, to further define the most likely origin of the strains, we compared the TRS in the IGR between I73R and I329L with 51 genotype II reference strains from Europe and Asian countries isolated between 2007 and 2022. The presence of TRS insertions in IGR between the I73R and I329L genes was first described in 1992 [52] and subsequently analyzed as an effective ASFV genome marker to discriminate closely circulating ASFVs from Eastern Europe [53], Russia [54] and Italy [33]. Recent ASF outbreaks in Italy were caused by ASFV strains of the IGR variant II, identical to the strains which have frequently occurred in wild boar and domestic pigs in EU countries since ASFV’s introduction into EU in 2014 and Asian countries after 2018.

Overall, these results confirm the remarkable genetic stability of the ASFV genotype II. It is not unexpected that the sequences of 632/AL/2022 and 2802/AL/2022 have extremely high homology with sequences, which identify to genotype II strains over a wide geographic area. Particularly, the mean genetic diversity is very low among the ASFV genotype II strains collected in Italy and the other genotype II strains identified in different countries from 2007 to date. Generally, in countries where multiple mechanisms of ASF transmission (mixed sylvatic and domestic cycles) play crucial roles in disease epidemiology, higher levels of variation are observed among viruses. In Italy and other European countries, ASFV strains have not been circulating in a sylvatic cycle as they have to date within the wild boar population. Despite the limited data, the potential involvement of ticks in the transmission of ASFV can be excluded [55].

Presently, given the high sequence similarity, it is impossible to trace the exact geographic origin of the virus to Northwest Italy in January 2022 at a country level. Moreover, the full-length sequences available in the NCBI are not completely representative of all affected territories. Therefore, the exact route and time of ASFV’s introduction remain unknown.

## Figures and Tables

**Figure 1 pathogens-12-00372-f001:**
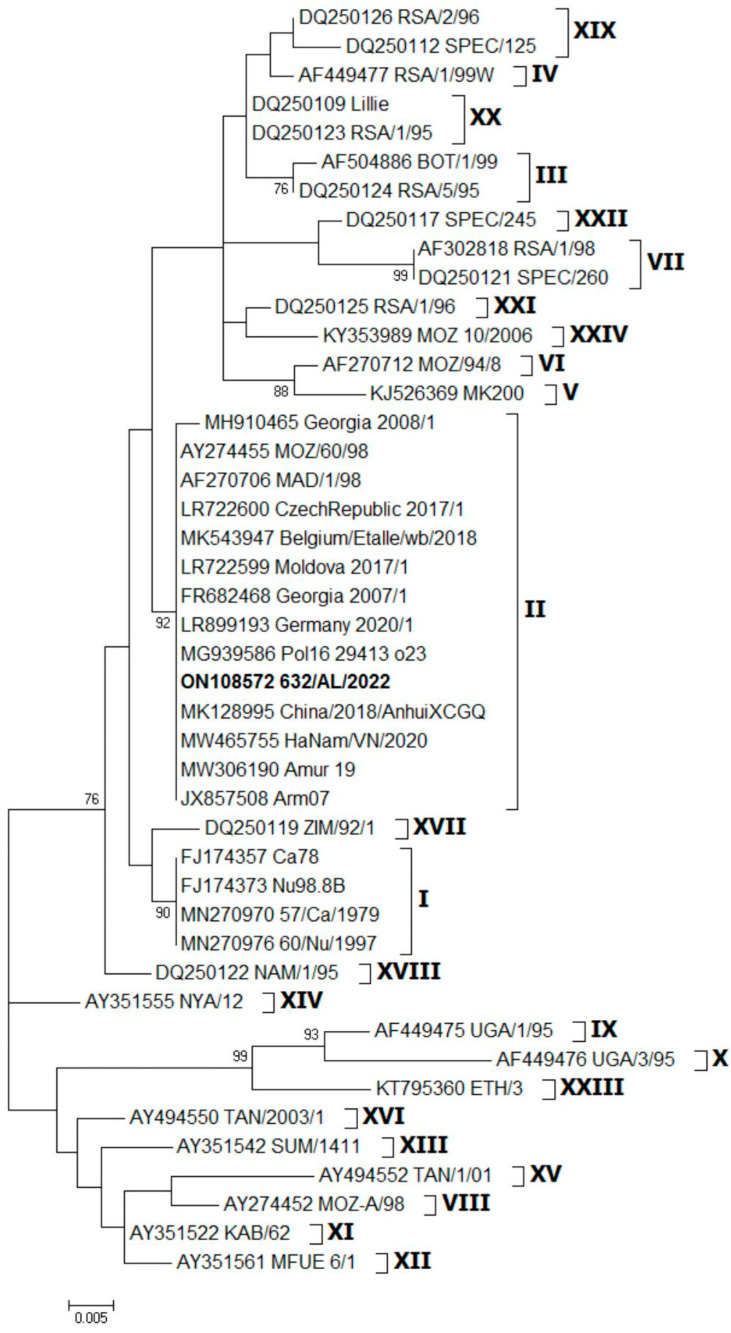
Phylogenetic tree reconstruction based on the C-terminal region of the B646L gene (p72). The sample ON108572 632/AL/2022 characterized in this study is shown in bold. Bootstrap values > 70 are indicated at their respective nodes. Bars indicate the number of nucleotide substitutions per site.

**Figure 2 pathogens-12-00372-f002:**
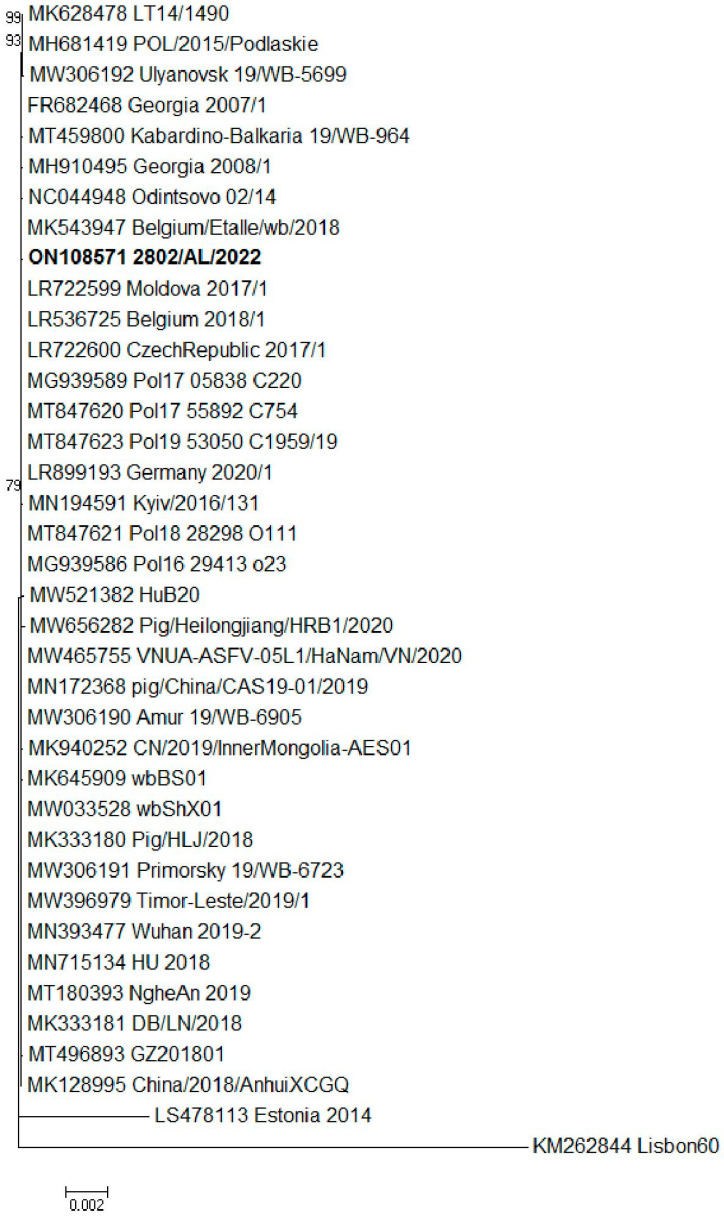
Maximum likelihood (ML) phylogenetic tree of 36 ASFV complete genome sequences. The sample ON108571 2802/AL/2022 characterized in this study is shown in bold. Bootstrap values > 70 are indicated at their respective nodes. Bars indicate the number of nucleotide substitutions per site.

**Figure 3 pathogens-12-00372-f003:**
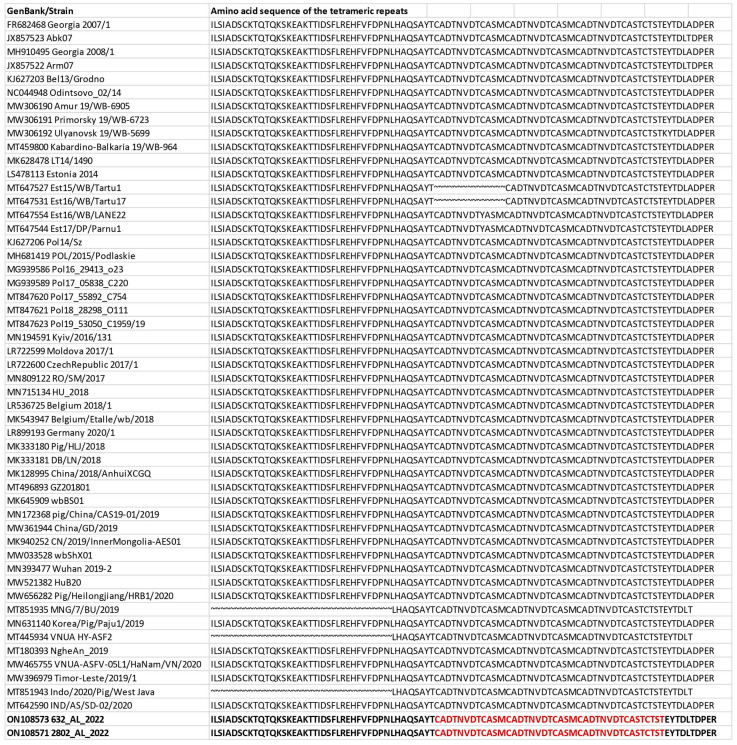
Amino acid sequence of the tetrameric repeats that constitute the central variable region (CVR) of the B602L gene identified in Italy. The single letters refer to the code of each tetrameric repeat: B = CADT; N = NVDT/NVGT; D = CASM; A = CAST; L = CTST; H = NEDT; P = NADT; S = SAST; O = NASI; F = NAST; Q = NADI; V = NANT; M = NANI; T = NVNT; C = GAST; K = CANT [17,22,29].

**Figure 4 pathogens-12-00372-f004:**
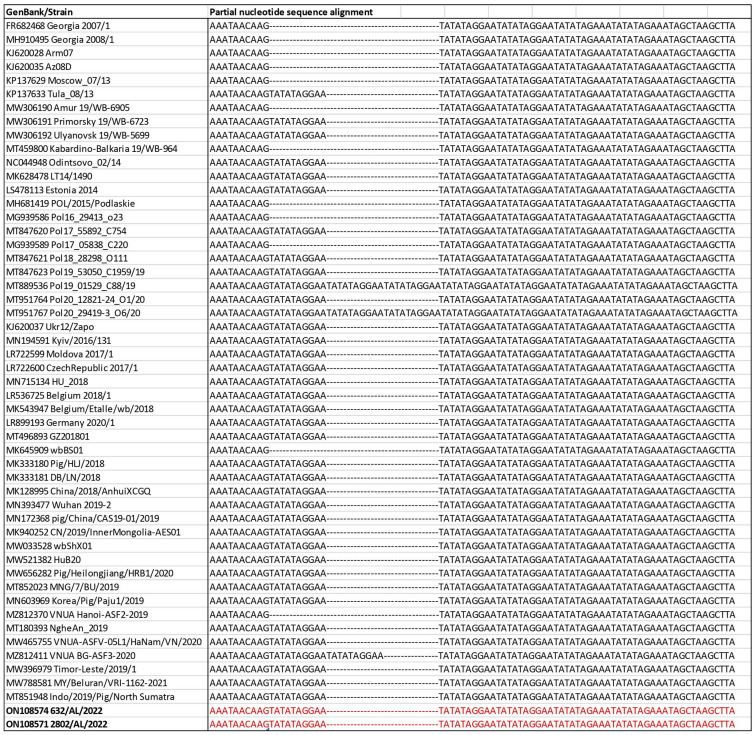
Nucleotide sequence alignment of the partial intergenic region between I73R and I329L from African swine fever genotype II viruses. Viruses characterized in this study are indicated in bold.

## Data Availability

The partial and full sequences generated in this study have been deposited in the NCBI GenBank database www.ncbi.nlm.nih.gov/ accessed on 29 March 2022.

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
