# Peer review of "Molecular Characterization of the First African Swine Fever Virus Genotype II Strains Identified from Mainland Italy, 2022"

_pathogens, 2023, doi:10.3390/pathogens12030372_

Round 1
Reviewer 1 Report
This manuscript tried to present the sequences and phylogenetic relationships of the ASFV isolates from the outbreaks in mainland Italy. However, the current version of this manuscript showed poor organization and language quality. In addition, this manuscript needs further analysis of the sequences, present the results in sufficient detail. Overall, the current version of this manuscript is not good enough to be considered as a research article for Pathogens. I recommend the authors revise it to a short communication.
Major Comments:
- Title: change “First Isolates” to “The first isolate”.
- Reorganization of the Introduction is needed.
- Reorganization of the results is needed. It can be separated with several parts.
- I advise the authors to find a native English speaker to proofread the manuscript.
Minor revisions:
- Family and Genus should be italic: Such as family Asfaviridae, genus Asfivirus.
- Page 2 lines 68,77, 78, 83, 86, 90, 91: “)” should be deleted.
- Page 4 line 181: please add the information in “()”.
Author Response
"Please see attachment"

Reviewer 2 Report
ü Reviewer comment
=> Reviewer comment
General comments
Sequence analysis can be used to distinguish between evolutionarily similar isolates. The latest advances in whole-genome sequencing are used to perform comprehensive genotyping and obtain data that are essential for elucidating the biology and genetic characteristics of ASFV. Using next-generation sequencing (NGS) and bioinformatics analysis for the detection. Sanger sequencing and NGS to investigate the epidemiological links between ASFV strains is important for scientific society. Overall, the results confirm the remarkable genetic stability of ASFV genotype II high homology with the sequences of genotype II strains over a wide geographic area. Generally, in countries where multiple mechanisms of ASF transmission (mixed sylvatic and domestic cycles) play crucial roles in the epidemiology of the disease, higher levels of variation are observed among viruses. Although, ASF V strains have not been circulating in a sylvatic cycle as the virus circulates primarily within a domestic pig-to-pig cycle and in the wild boar population in European countries, potential ticks transmission of ASFV can be possible, given the high sequence similarity. Also, it is difficult to determine the exact geographic origin. Therefore, the exact route and time of the introduction remain unknown.
Abbreviation, please indicate full words for first use at the text of article.
References should be based on journal constriction.
# Reviewer comment #
o General and specific comment inserts along manuscript
o Title need improved
o Ethical approval should present
o Introduction please mention problem
o Abbreviation, please indicate full words for first use at the text of article.
o Update the references very carefully.
o References should be based on journal constriction.
Author Response
"Please see the attachment."

Reviewer 3 Report
Review report
Article „Molecular Characterization of First Isolates of African Swine 2 Fever Virus Genotype II in Italy“, ID: pathogens-2146803
African swine fever (ASF) is a highly contagious viral disease of domestic and wild pigs, whose mortality rate can reach 100%. There are neither vaccines nor cures. It is not a danger to human health, but it is responsible for devastating effects on pig populations and the farming economy and it has become a major crisis for the pork industry in recent years. The authors described the molecular characterization of the first ASF isolates detected in wild boar population reported in mainland Italy in 2022 in order to try to trace the geographic origin of the virus.
Abstract
Line 22- please rephrase “by partial sequencing (Sanger) of a fragment of the B646L gene, a fragment 22
of the B602L gene” into by partial sequencing (Sanger) of the B646L gene fragment, B602L gene fragment
Lines 35-37. Please rephrase the following sentence: “Moreover, the full length sequences available in NCBI are not completely representative of certain of the affected territories.”
Keywords
Maybe to add: ASF outbreak, mainland Italy, wild boar population
1. Introduction
Line 46. remained instead of remain
Line 49. used as swine feed instead of used to feed pigs
Line 68. Remove parenthesis after 400 bp
Lines 76., 77., 78., 80., 83., 86., 88., 90., 91., Remove parenthesis
2. Materials and Methods
Lines 112- 114. Maybe to rephrase sentences into: During the first two weeks of January 2022 when ASF outbreak in Italy (Piedmont region) occurred, wild boar bone marrow and spleen samples were collected from the affected area.
Line 162. Please use number for citation (Granberg et al., 2016 is listed as number 37 in the chapter references)
3. Results
Line 172. Of a B646L gene fragment, B602L gene fragment (as in abstract)
Figure 1.&2. It would be good if the authors would indicate the host (domestic pig, wild boar) of the reference sequences retrieved from GenBank (next to Acc. no on the phylogenetic tree or as separate table)
4. Discussion
Since the exact route and time of ASF introduction remains unknown (You excluded the potential involvement of ticks), could You please comment could the swine trade or the wild boar migrations be responsible for the recent ASF outbreak in Italy?
Author Response
"Please see the attachment."

Round 2
Reviewer 1 Report
okay
Author Response
[February 20th, 2023]
Dear Reviewer,
I wish to re-submit the manuscript no.2146803, for publication in the journal entitled, “Molecular Characterization of the First African Swine Fever Virus Genotype II strains identified from Mainland Italy, 2022.” for consideration as an article. The paper was co-authored by Dondo Alessandro, Cammà Cesare, Masoero Loretta, Torresi Claudia, Marcacci Maurilia, Zoppi Simona, Curini Valentina, Antonio Rinaldi, Rossi Elisabetta, Casciari Cristina, Pela Michela, Pellegrini Claudia, Iscaro Carmen and Feliziani Francesco.
Reviewer 1' comments:
……… The most important comment is that the references for the primers and assays used for the gene-level characterisation should be added to ensure that the methodology is repeatable……………
We really appreciate the reviewer 1 comments. We introduce the reference.